# Text-Guided Diffusion Based Ambiguous Medical Image Segmentation

## Abstract

Medical image segmentation often suffers from ambiguity due to unclear boundaries, expert inconsistencies, and varying interpretation standards. Traditional segmentation models produce single deterministic outputs, failing to capture this uncertainty and the range of plausible interpretations in such cases. In this work, we introduce AmbiguousTextDiff, a novel text-guided diffusion model that generates diverse and plausible segmentation proposals reflecting the ambiguity observed in medical imaging. By combining the strengths of text-conditional diffusion models with ambiguity-aware training, our approach generates multiple valid segmentations for a single input image. We use descriptive text prompts incorporating anatomical, morphological, and diagnostic attributes as conditioning signals to guide segmentation. These prompts are generated by extracting clinical metadata from two diverse sources: the LIDC-IDRI lung nodule dataset (e.g., texture, spiculation, malignancy) and the IMA++ skin lesion dataset (e.g., anatomical site, pathology). This text-based conditioning improves both the controllability and clinical relevance of the model's outputs, aligning them more closely with radiologist interpretation. Extensive evaluations and ablations on both datasets demonstrate that AmbiguousTextDiff achieves superior performance across Combined Sensitivity, Diversity Agreement, Generalized Energy Distance (GED), and Collective Insight (CI) Score. Our results highlight the value of text-guided diffusion for ambiguity-aware segmentation across multiple imaging modalities and establish a new direction for controllable and interpretable medical image analysis.

## 1 Introduction

Medical image segmentation plays an important role in modern medical image analysis, serving as an essential step in numerous clinical applications such as disease diagnosis, treatment planning, surgical guidance, and patient monitoring. Accurately identifying organs and diseased regions is essential for understanding and analyzing medical images. However, in real-world practice, these images often carry inherent uncertainties, making it difficult to achieve consistent segmentation results that all doctors and experts agree upon.

Several factors contribute to these ambiguities:

- **Blurred anatomical boundaries:** In many medical images, especially MRI or ultrasound, the boundaries of organs or abnormal areas are often unclear and blend with nearby tissues, making it difficult to accurately identify them.

- **Inherent limitations in imaging modalities:** Factors such as unwanted marks (artifacts), random noise, blurriness from movement, low contrast, and poor image quality make it harder to understand and trust what we see in medical images. A sample is shown in Figure 1.

- **Subjective interpretation by medical experts:** Inter-observer variability can occur when multiple radiologists or clinicians annotate the same image differently because of differences in training, clinical focus, or personal experience. This is well-documented in datasets such as LIDC-IDRI (Armato III et al., 2011).

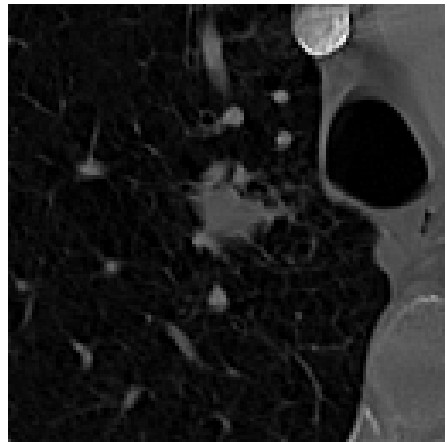

Figure 1: Text annotation corresponding to the image: `The annotation id is 128. The subtlety is 5. The internalStructure is 1. The calcification is 6. The sphericity is 5. The margin is 4. The lobulation is 1. The spiculation is 5. The texture is 4. The malignancy is 4. The Subtlety is Obvious. The InternalStructure is Soft Tissue. The Calcification is Absent. The Sphericity is Round. The Margin is Near Sharp. The Lobulation is No Lobulation. The Spiculation is Marked Spiculation. The Texture is Solid/Mixed. The Malignancy is Moderately Suspicious. The diameter mm is 31.92. The volume mm`$^3$` is 11568.45. The slice indices is [101-113]. The bbox is (slice(264,315), slice(173,217), slice(101,114)). Average malignancy (meta): 5.00. Is cancer (any slice): True. Is clean (all slices): False. Diameter: 20.6846910857402 mm. Malignancy level: 5.0. Malignancy (nodule): 1.0.` Although there are descriptions of the concerned region, the requirement is to identify the segmentation boundaries.

- **Contextual dependency on clinical priorities:** Depending on the medical setting, such as screening vs treatment monitoring or pediatric versus adult populations, different criteria may be deemed relevant or unusual.

In our own experiments with the LIDC-IDRI dataset, we frequently saw radiologists' annotations for the same nodule vary dramatically. These disagreements span everything from high-level labels—like 'suspicious', 'benign', or even 'non-nodule'—to the fine-grained pixel-level boundaries themselves. This reflects genuine clinical ambiguity rather than annotation noise. By chasing a single 'ground truth' output, current models risk ignoring the multi-modality that defines real-world medical imaging. **Our research asks a fundamental question: can a model learn to represent the full spectrum of expert disagreement, instead of collapsing it into a single, misleading average?**

Recent advances have explored using diffusion models for segmentation (Wolleb et al., 2021; Wu et al., 2022), but they often fall short in one of two ways. **While recent works like TextDiff (Feng, 2024) utilize text for label-efficient deterministic segmentation, they typically yield a single, deterministic output, failing to capture the necessary output variability for ambiguous cases. In contrast, our work is the first to leverage text prompts to explicitly model the distribution of clinical ambiguity.** Conversely, conditional diffusion models that generate multiple segmentations, such as CIMD (Rahman et al., 2023), often lack strong semantic control, especially when visual information is insufficient or unclear. Our work addresses both limitations by introducing a framework that is both text-guided for semantic control and inherently stochastic to model the full distribution of plausible clinical interpretations. Crucially, we also demonstrate the importance of rigorous evaluation on complete, unfiltered datasets, revealing that the performance of prior methods may be overestimated when tested on curated subsets.

## 1.1 Our Contributions

To overcome this significant constraint, we present **AmbiguousTextDiff**, a novel framework in this study. By producing *multiple plausible segmentation masks* that are conditioned on both the image and a textual description of the ambiguity, *AmbiguousTextDiff*, a **text-guided diffusion-based generative model**, explicitly captures the uncertainty and diversity in medical image segmentation. Inspired by recent advancements in diffusion models and multi-modal learning, our approach introduces a new direction for modeling,

exploring, and interpreting segmentation ambiguity. To the best of our knowledge, this is among the first works to explicitly combine text-guided diffusion modeling with ambiguity-aware medical image segmentation.

**Our key contributions include:**

- **A Unified Framework for Text-Conditional Diffusion:** We create a new diffusion-based architecture that incorporates textual descriptions as conditioning inputs as shown in Figure 2, allowing the model to produce segmentations that represent ambiguous features learned during training or described by medical experts.

- **Novel Training Strategy:** We integrate domain-informed text prompts, simulated ambiguous regions, and diverse ground-truth annotations. Instead of predicting just one average result, the model learns to produce a range of possible segmentations.

- **Diversity and Quality-Centric Evaluation Metrics:** To evaluate the diversity, plausibility, and clinical validity of the generated results, we use specialized metrics like Generalized Energy Distance (GED) (Selvan et al., 2020), Collective Insight (CI) score, and Diversity Agreement (Rahman et al., 2023). Since multiple valid outputs are possible, we do not rely on standard single-output overlap metrics like Dice or IoU.

- **Comprehensive Experimental Validation:** We perform comprehensive experiments on two distinct modalities: lung CT LIDC-IDRI dataset(grayscale) (Armato III et al., 2011) and skin dermoscopy (color) (Abhishek et al., 2025) under simulated ambiguity. Our results show that AmbiguousTextDiff outperforms existing deterministic and stochastic segmentation methods in capturing clinically important variations.

It is important to distinguish between the descriptive metadata used in our text prompts and the segmentation task itself. Metadata like nodule diameter, texture, or even a bounding box provides a high-level, coarse description of a region of interest. What it does not provide is the pixel-level detail needed for clinical applications like volumetric analysis, radiotherapy planning, or precise morphological assessment. Our model uses this descriptive information as context, guiding it to generate fine-grained boundary segmentations. In essence, the metadata is not meant to replace pixel-level annotations but to help resolve ambiguity at the pixel scale, mirroring how a radiologist uses clinical context to interpret unclear boundaries.

## 2 Related Work

### 2.1 Medical Image Segmentation

Medical image segmentation plays a key role in computer-aided diagnosis and treatment planning. Over the years, it has evolved from early methods like thresholding, edge detection, and region growing, which often struggled with noise and generalization to much more powerful deep learning approaches. A major breakthrough came with convolutional neural networks (CNNs), especially the U-Net architecture (Ronneberger et al., 2015), whose encoder-decoder design and skip connections help preserve spatial details essential for accurate segmentation. U-Net and its many variants have since become standard tools across segmentation tasks. More recently, researchers have pushed this further by blending CNNs with transformers. For example, TransUNet (Chen et al., 2021) enhances the classic U-Net by adding transformer modules that can capture long-range dependencies, particularly useful for understanding complex anatomical structures. On the other hand, UNETR (Hatamizadeh et al., 2022) places transformers right at the start of the encoder path, allowing the model to process 3D medical data more effectively by leveraging global contextual information.

### 2.2 Ambiguity in Medical Segmentation

One of the biggest ongoing challenges in medical image segmentation is dealing with ambiguity: a common issue that arises from low image quality, overlapping anatomical structures, and differences in expert opinions. Even with the progress in segmentation models, this uncertainty remains hard to tackle. Multiple studies have

highlighted the need to model such ambiguity rather than ignore it (Kohl et al., 2018). Instead of producing a single deterministic result, researchers have proposed probabilistic models that can learn from several expert annotations and provide a variety of plausible segmentations. This unpredictability is explicitly captured by architectures such as the probabilistic U-Net (Kohl et al., 2018), which reflects the actual uncertainty observed in clinical practice. A more structured and comprehensible method of modeling uncertainty has been provided by more recent attempts to go one step further with hierarchical models that arrange ambiguity at several levels (Kohl et al., 2019; Baumgartner et al., 2019). By better simulating radiologist's reasoning in ambiguous situations, these methods hope to bring automated segmentations closer to expert-level decision-making.

### 2.3 Diffusion Models

Diffusion models have recently gained attention as powerful tools for generating images, often setting new benchmarks in image synthesis. They started with denoising diffusion probabilistic models (DDPMs) (Sohl-Dickstein et al., 2015), which were originally introduced for natural image synthesis, which laid the foundation for many later improvements (Ho et al., 2020). One major breakthrough was latent diffusion models like Stable Diffusion (Rombach et al., 2022), which shift the denoising process to a lower-dimensional latent space instead of the raw image space. This not only boosts efficiency but also enables the generation of high-quality, high-resolution images guided by simple text prompts. These advances are particularly promising for medical applications, where generating or processing large, high-dimensional data (like 3D scans) requires both accuracy and efficiency.

### 2.4 Text-Guided Image Generation

Combining text with generative models has made huge strides in recent years. Models like DALL·E, Stable Diffusion (Rombach et al., 2022), and Imagen have shown that it's possible to create realistic, high-quality images directly from text prompts; making image generation more intuitive and flexible, especially in cases where traditional labels or bounding boxes fall short. More recently, these text-guided generation techniques have started making their way into medical imaging, where they're being used for tasks like segmentation and image synthesis (Feng, 2024). Text-guided generation is being explored to incorporate expert knowledge like radiology reports or clinical annotations into the generation process.

## 3 Methodology

Our proposed method, AmbiguousTextDiff, leverages a text-guided diffusion model to address the inherent ambiguity in medical image segmentation. Instead of producing a single deterministic output, we aim to model the conditional distribution of plausible segmentation masks $p(\boldsymbol{y}|\boldsymbol{x}, \boldsymbol{c})$ given an input image $\boldsymbol{x}$ and a descriptive text prompt $\boldsymbol{c}$. This allows us to generate a diverse ensemble of segmentation proposals $\{\boldsymbol{y}^{(1)}, \boldsymbol{y}^{(2)}, \ldots, \boldsymbol{y}^{(N)}\}$ that reflects the range of valid interpretations present in ambiguous cases, such as those arising from inter-observer variability among clinical experts.

### 3.1 Denoising Diffusion Probabilistic Models Preliminaries

Denoising Diffusion Probabilistic Models (DDPMs) are a class of generative models consisting of two processes: a fixed forward diffusion process and a learned reverse denoising process (Ho et al., 2020).

**Forward Process (Diffusion).** The forward process gradually adds Gaussian noise to an initial data sample $\boldsymbol{y}_0$ (such as a clean segmentation mask) over $T$ discrete timesteps. This is done step-by-step using a Markov chain, where each noisy version $\boldsymbol{y}_t$ is sampled from the previous one $\boldsymbol{y}_{t-1}$:

$$q(\boldsymbol{y}_t|\boldsymbol{y}_{t-1}) = \mathcal{N}(\boldsymbol{y}_t; \sqrt{1-\beta_t}\boldsymbol{y}_{t-1}, \beta_t\boldsymbol{I}), \tag{1}$$

with $\{\beta_t\}_{t=1}^T$ defining the noise schedule. A useful property of this setup is that we can directly sample $\boldsymbol{y}_t$ from the original input $\boldsymbol{y}_0$, without simulating all intermediate steps—using:

$$q(\boldsymbol{y}_t|\boldsymbol{y}_0) = \mathcal{N}(\boldsymbol{y}_t; \sqrt{\bar{\alpha}_t}\boldsymbol{y}_0, (1-\bar{\alpha}_t)\boldsymbol{I}), \tag{2}$$

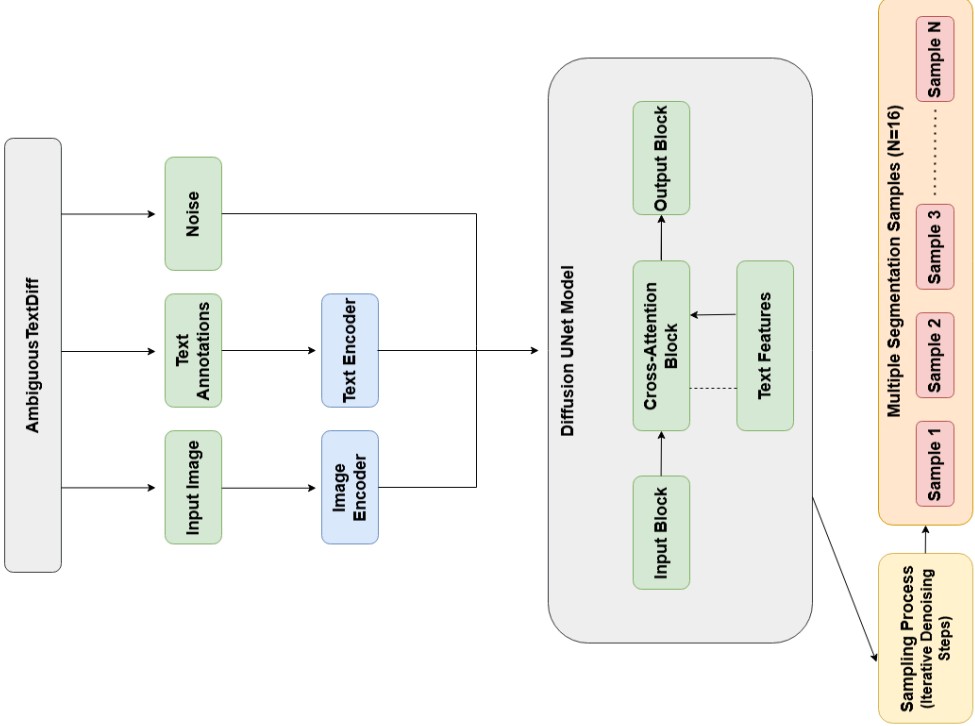

Figure 2: An overview of the *AmbiguousTextDiff* architecture, which takes an input image, text annotations, and noise, feeding the encoded features into a diffusion U-Net where cross-attention guides the generation of multiple plausible segmentation outputs.

where $\alpha_t = 1 - \beta_t$ and $\bar{\alpha}_t = \prod_{s=1}^{t} \alpha_t$. This makes it efficient to generate noisy versions of the original data at any point in the diffusion process.

**Reverse Process (Denoising).** The reverse process aims to reconstruct the original data by gradually removing noise, starting from a completely noisy input $\boldsymbol{y}_T \sim \mathcal{N}(\boldsymbol{0}, \boldsymbol{I})$. This denoising is learned through a neural network parameterized by $\boldsymbol{\theta}$, which approximates the true reverse distribution $p(\boldsymbol{y}_{t-1}|\boldsymbol{y}_t)$ at each step:

$$p_{\boldsymbol{\theta}}(\boldsymbol{y}_{t-1}|\boldsymbol{y}_t) = \mathcal{N}(\boldsymbol{y}_{t-1}; \boldsymbol{\mu}_{\boldsymbol{\theta}}(\boldsymbol{y}_t, t), \boldsymbol{\Sigma}_{\boldsymbol{\theta}}(\boldsymbol{y}_t, t)). \tag{3}$$

Instead of directly predicting the denoised data, the model is commonly trained to estimate the noise $\epsilon$ that was originally added making it easier to learn the denoising process and improve training stability.

## 3.2 AmbiguousTextDiff Architecture

Our AmbiguousTextDiff model, depicted in **Figure 2**, is a conditional diffusion framework designed for text-guided ambiguous segmentation. It combines information from two sources: the input image, which acts as a fixed condition, and a text prompt that provides semantic guidance. The architecture consists of a visual pathway to process the image, a text encoder to interpret the prompt, and a conditional U-Net denoiser at its core. These components are connected using cross-attention, which allows the model to effectively align and fuse the multi-modal information during the denoising process.

**Input Representation and Conditioning.** Our model takes a medical image $\boldsymbol{x}$, normalized to the range [-1, 1], as a fixed condition throughout the diffusion process. A crucial design choice is that the diffusion process applies **only to the segmentation mask**, not the image itself. At each forward timestep $t$, Gaussian

noise is added only to the mask channel $\boldsymbol{y}_0$ to produce a noisy mask $\boldsymbol{y}_t$. The U-Net denoiser receives the input image and the noisy mask concatenated along the channel dimension, forming the input tensor $\boldsymbol{z}_t = [\boldsymbol{x}, \boldsymbol{y}_t]$. This architecture ensures that the model learns the distribution of valid segmentations conditioned on a static visual context.

**Text-Guided U-Net Denoiser.** The core of our model is a U-Net denoiser adapted to incorporate text guidance. It follows a standard encoder-decoder structure with skip connections, operating at multiple spatial resolutions (32, 16, and 8). To enhance its representational capacity, we integrate residual blocks and self-attention modules at each resolution level. This enables the network to model complex spatial dependencies while being guided by both the image features and time embeddings, which encode the current diffusion timestep $t$.

**Cross-Attention Mechanism.** To integrate textual guidance, we employ cross-attention layers at multiple resolution levels of the U-Net, serving as a bridge between textual and visual modalities (**Figure 3a**). The text prompts are first encoded into 768-dimensional embedding vectors $\boldsymbol{c}$ by a pre-trained Bio_ClinicalBERT model (Alsentzer et al., 2019). These embeddings are projected to form the key ($\boldsymbol{K}$) and value ($\boldsymbol{V}$) matrices, while the U-Net's intermediate feature maps provide the query ($\boldsymbol{Q}$). The attention operation is defined as:

$$\text{Attention}(\boldsymbol{Q}, \boldsymbol{K}, \boldsymbol{V}) = \text{softmax}\left(\frac{\boldsymbol{Q}\boldsymbol{K}^T}{\sqrt{d_k}}\right)\boldsymbol{V}, \tag{4}$$

where $d_k$ is the dimensionality of the key vectors. We inject the text embeddings at three resolution levels (32, 16, and 8) in both the encoder and decoder paths. At each level, the feature maps act as queries and the projected text embeddings supply keys and values, ensuring that semantic guidance consistently modulates the denoising process across different feature scales. This design channels the model's stochasticity toward clinically meaningful interpretations.

### 3.3 Ambiguity-Aware Training

During training, the model learns to predict the noise that was added to the segmentation mask in the forward diffusion process. This prediction is guided by three inputs: the medical image, a text prompt describing the expected outcome, and the current diffusion timestep. By conditioning on all three, the model learns to generate segmentations that reflect both the visual context and the semantic variations captured in the text, while accounting for ambiguity in the data. The overall process is visualized in **Figure 3b**.

**Hybrid Loss Function.** To train the model to be both accurate and ambiguity-aware, we employ a hybrid loss function that combines a standard denoising objective with a latent space regularization term:

$$\mathcal{L}_{\text{total}} = \mathcal{L}_{\text{mse}} + \lambda \mathcal{L}_{\text{kl}}. \tag{5}$$

**The weighting factor $\lambda$ is set to $0.001$ to balance the numerical scales. Since pixel-space MSE typically has a larger magnitude than latent KL divergence, this scaling ensures MSE remains the dominant gradient signal for structure, while the KL term acts as a regularizer without causing training instability or dominant fluctuations.**

The primary component, $\mathcal{L}_{\text{mse}}$, is the standard denoising score matching objective from DDPMs (Ho et al., 2020), implemented as the Mean Squared Error (MSE) between the true and predicted noise in pixel space:

$$\mathcal{L}_{\text{mse}} = \mathbb{E}_{\boldsymbol{y}_0, \boldsymbol{c}, \epsilon, t}\left[\|\epsilon - \epsilon_{\boldsymbol{\theta}}(\boldsymbol{z}_t, \boldsymbol{c}, t)\|^2\right], \tag{6}$$

where $\boldsymbol{z}_t = [\boldsymbol{x}, \boldsymbol{y}_t]$ is the concatenation of the input image and the noisy mask, and $\epsilon_{\boldsymbol{\theta}}$ is the noise predicted by our U-Net. To explicitly model the distribution of ambiguous annotations, we introduce a KL-divergence term, $\mathcal{L}_{\text{KL}}$, that operates in a learned latent space, inspired by the Probabilistic U-Net (Kohl et al., 2018). This term is computed using two lightweight convolutional networks: a **posterior network** $q_{\phi}(z \mid \boldsymbol{x}, \boldsymbol{y}_0)$ and a **prior network** $p_{\psi}(z \mid \boldsymbol{x}, \hat{\boldsymbol{y}}_0)$, where $\hat{\boldsymbol{y}}_0$ is the U-Net's prediction of the clean mask. The posterior network encodes the ground-truth mask $\boldsymbol{y}_0$ into a latent distribution, while the prior network encodes the

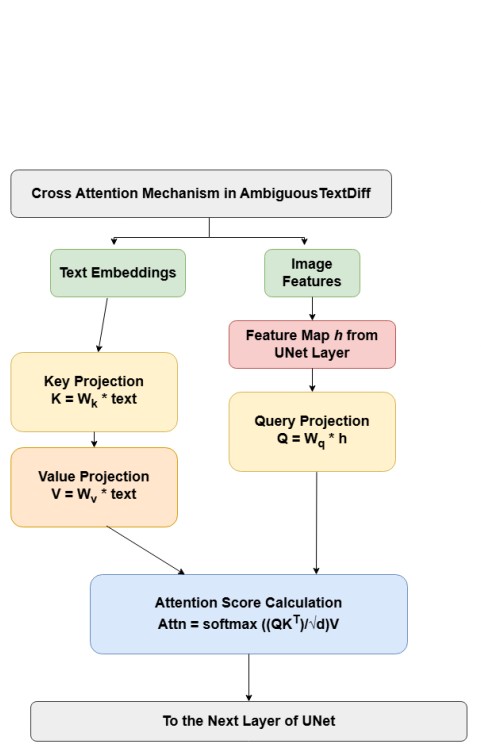 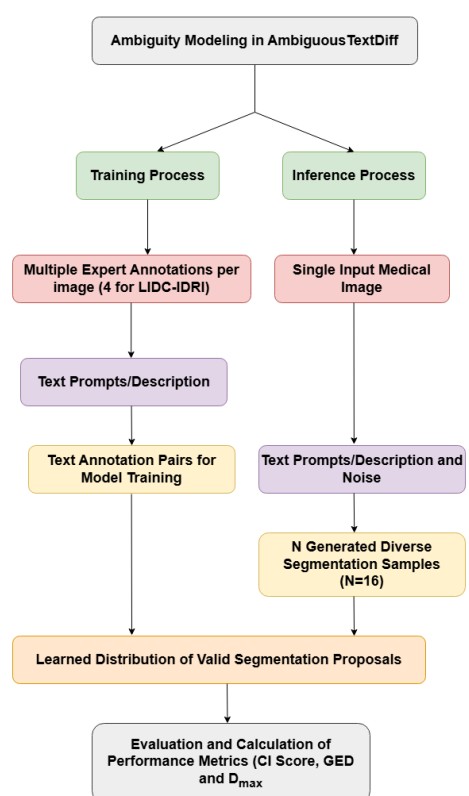

(a) Cross Attention Mechanism in *AmbiguousTextD-iff*, where text embeddings guide image features via key-value-query projections.

(b) Ambiguity Modeling in *AmbiguousTextDiff*, illustrating the training and inference processes with multi-annotator inputs and diverse generation.

Figure 3: Key components in *AmbiguousTextDiff*: attention mechanism and ambiguity-aware generation.

model's own prediction. The KL-divergence then forces the model's predictions to be encodable into a latent distribution that is consistent with the ground truth:

$$\mathcal{L}_{\mathrm{KL}} = \mathrm{KL}(q_{\boldsymbol{\phi}}(z \mid \boldsymbol{x}, \boldsymbol{y}_0) \parallel p_{\boldsymbol{\psi}}(z \mid \boldsymbol{x}, \hat{\boldsymbol{y}}_0)). \tag{7}$$

This latent regularization encourages the U-Net to generate segmentations that lie on the manifold of plausible expert annotations. The weighting factor $\lambda$ is a crucial hyperparameter, which we detail in our implementation details (Appendix A).

**Training Process.** The model is trained through an iterative process. In each training step, we sample a mini-batch consisting of image, ground-truth mask, and text prompt triplets $(\boldsymbol{x}, \boldsymbol{y}_0, \boldsymbol{c})$ from the dataset. A random timestep $t$ is selected uniformly, and Gaussian noise $\epsilon$ is added to the ground-truth mask to generate a noisy version $\boldsymbol{y}_t$, following the forward diffusion schedule. The U-Net denoiser then takes the noisy input $\boldsymbol{y}_t$ concatenated with the original image $\boldsymbol{x}$, along with the current timestep $t$ and the corresponding text embedding $\boldsymbol{c}$. Its goal is to predict the original noise, denoted as $\epsilon_{\boldsymbol{\theta}}$. Using this prediction, we compute the hybrid loss $\mathcal{L}_{\mathrm{total}}$, and update the model parameters $\boldsymbol{\theta}$ using gradient descent. This training cycle is repeated until the model successfully learns to generate meaningful segmentations that reflect both the image content and the semantic guidance from the text.

### 3.4 Inference for Diverse Segmentation Generation

During inference, AmbiguousTextDiff takes advantage of its generative nature to produce a diverse set of segmentation masks, each representing a different but plausible interpretation of the input image $\boldsymbol{x}$. Specifically, we generate 16 unique segmentation proposals by introducing diversity through two key mechanisms: variation in text prompts and randomness in sampling. To enable this, we created an annotation CSV by extracting and organizing metadata and descriptive information from the dataset. This CSV provides a set of diverse text prompts designed to reflect different semantic perspectives and guide the model toward varied interpretations during segmentation.

1. **Text Variation:** We use these curated text prompts $\boldsymbol{c}$ to guide the model toward different plausible anatomical or semantic interpretations of the image.

2. **Stochastic Initialization:** For each prompt, the reverse diffusion process is initialized with a different random noise sample $\boldsymbol{y}_T \sim \mathcal{N}(\boldsymbol{0}, \boldsymbol{I})$. The stochastic nature of this process ensures that even with the same text input, the model can produce distinct but valid segmentation outputs.

This dual-source diversity allows the model to capture the real-world ambiguity inherent in medical imaging tasks, producing a spectrum of clinically plausible segmentations. The generation process unfolds over $T$ reverse diffusion steps, starting from an initial noise sample $\boldsymbol{y}_T$. At each step $t$, the model predicts a slightly less noisy version of the mask, $\boldsymbol{y}_{t-1}$, based on the current noisy input $\boldsymbol{y}_t$, the original image $\boldsymbol{x}$, and the selected text prompt $\boldsymbol{c}$. Through this iterative process, the model gradually refines the noisy input, eventually producing a coherent segmentation mask that reflects both the visual content of the image and the semantic guidance from the text.

## 4 Experiments

### 4.1 Datasets and Preprocessing

**LIDC-IDRI (Lung CT):** We conduct our experiments on the LIDC-IDRI dataset (Armato III et al., 2011), which is widely considered the gold-standard benchmark for evaluating segmentation ambiguity due to its unique multi-radiologist annotations. It provides thoracic CT scans with detailed lung nodule annotations. While our method is conceptually generalizable, we focus our evaluation on LIDC-IDRI as it provides the dense, real-world inter-observer variability essential for validating our core contributions in ambiguity modeling. We use the Kaggle-hosted version (ZhangWeiLed (on Kaggle), 2025) of this dataset, which offers preprocessed scans and annotations derived from the original collection released by the Lung Image Database Consortium. With 1012 subjects, LIDC-IDRI stands as one of the most comprehensive resources for computer-aided lung cancer diagnosis. Each scan in the dataset underwent a rigorous two-stage annotation process involving four board-certified thoracic radiologists. In the initial (blinded) phase, each radiologist independently annotated the CT slices, classifying findings into three categories: **nodules $\geq$ 3 mm**, **nodules $\leq$ 3 mm**, and **non-nodules $\geq$ 3 mm**. In the subsequent (unblinded) phase, they reviewed each other's anonymized annotations and refined their decisions, resulting in a unique multi-reader consensus. Some slices, however, remain partially annotated or unannotated; we conservatively treat these cases as signal absence (black). Unlike previous works such as CIMD (Rahman et al., 2023), which focused on selected 2D lesion slices from the LIDC-IDRI dataset, our approach includes *all* nodules, both smaller and larger than 3 mm. While CIMD does not explicitly mention filtering by nodule size, their training and test sets (13,511 and 1,585 slices respectively) were constructed from lesion-centric views, likely favoring nodules with clearer or more consistent annotations. To ensure a fair and robust comparison, we also evaluate CIMD baseline method on the complete, unfiltered test set of 3,072 images. In contrast, we introduce descriptive textual guidance that allows our model to better handle a wider range of nodule sizes, especially small or ambiguous ones. We curate custom text annotations from radiology reports and scan metadata, explicitly highlighting attributes like nodule size. These descriptions help guide the diffusion model during both training and inference, enabling it to better capture clinically meaningful variations across the full size spectrum. Because our method does not rely on manual slice selection or size-based filtering, we are able to train on a

broader and more diverse dataset. Specifically, we consider all 3,072 nodules during evaluation, generating 16 diverse segmentation samples per test image. This strategy improves our ability to model uncertainty and ambiguity in a more comprehensive and clinically relevant manner. The codes and datasets are uploaded as supplementary material.

**IMA++ (Skin Dermoscopy):** To evaluate the generalizability of AmbiguousTextDiff across different imaging modalities, we evaluate on the IMA++ dataset (Abhishek et al., 2025). IMA++ is currently the largest publicly available multi-annotator skin lesion segmentation dataset. For our study, we focus on the multi-annotator subset containing 2,394 dermoscopic images, each possessing between 4 and 7 independent expert annotations, averaging 4.13 masks per image. This dataset contains 9,899 total masks. The ambiguity here arises from low contrast boundaries, the presence of clinical artifacts like hair or gel bubbles, and varying expert interpretations of the lesion perimeter.

**Data Partitioning and Preprocessing** For LIDC-IDRI, we follow the 80/20 subject-level split described previously. For IMA++, we adopt a 70/15/15 split by unique images to ensure zero leakage between sets. This results in a training set of 1,675 images (6,943 annotations), a validation set of 359 images (1,474 annotations), and a test set of 360 images (1,482 annotations). All images across both datasets were resized to $128 \times 128$ and normalized to the range $[-1, 1]$.

**Text Prompt Curation.** To provide meaningful semantic guidance, we curated descriptive text prompts by systematically extracting and structuring metadata from the LIDC-IDRI dataset for each of the 1012 subjects. The raw metadata for each nodule includes several radiologist-provided ratings (e.g., subtlety, sphericity, margin, texture, malignancy) on a 1-5 or 1-6 scale, along with corresponding descriptive labels (e.g., "Soft Tissue," "Marked Spiculation"). Our curation process involved converting these numerical and categorical ratings into natural language sentences using a predefined template. For instance, a 'spiculation' rating of 5 was translated to the phrase "The Spiculation is Marked Spiculation." This process was repeated for all available attributes, and the resulting sentences were concatenated to form a comprehensive descriptive paragraph for each nodule. During training and inference, the same subject-level prompt was used for all slices belonging to that subject. The long annotation shown in Figure 1 is an example of such a generated prompt. This structured approach ensures that the textual guidance is consistent, grounded in clinical data, and rich in semantic detail. We provide both numeric and text labels (e.g., "subtlety is 5" and "Subtlety is Obvious") as this redundancy was found empirically to create a more robust conditioning signal for the frozen text encoder.

**IMA++ Prompts:** For the skin lesion dataset, we curated highly detailed prompts by serializing the clinical and technical metadata into natural language. Each prompt includes the patient's demographics (age, sex), anatomical location, malignancy status, and a five-level hierarchical diagnosis ranging from general category to specific histological subtype (e.g., "Nevus, Dysplastic"). Additionally, we incorporate technical metadata such as the annotator ID and skill level to provide context for the specific segmentation style. A representative prompt generated by this process is: *"The patient is a 40-year-old female. The dermoscopic image shows a benign melanocytic lesion on the posterior torso, diagnosed as a Dysplastic Nevus. The annotation was performed by a novice annotator using a Lasso tool."* This rich semantic context directs the diffusion model to generate boundaries that align with both the pathological diagnosis and the specific interpretation style of the clinical metadata.

## 4.2 Implementation Details

We implemented our method in PyTorch and trained on 4 NVIDIA RTX 4090 GPUs. The core of our model is a U-Net denoiser with self- and cross-attention mechanisms, guided by textual embeddings from a frozen `Bio_ClinicalBERT` model (Alsentzer et al., 2019). We employed a 1000-step diffusion process and trained the model for 50,000 steps using the AdamW optimizer with an initial learning rate set to be $1 \times 10^{-4}$. To ensure stable training and better generalization, we utilized an Exponential Moving Average (EMA) of the model weights. A comprehensive list of all hyperparameters, including the noise schedule, architectural specifics, and optimizer details, is provided in the Technical Appendix in supplementary.

### 4.3 Baselines

We compare our approach against several existing state-of-the-art methods, including Probabilistic U-Net (Kohl et al., 2018), which generates diverse segmentations using a conditional VAE; PHiSeg (Baumgartner et al., 2019), a hierarchical probabilistic model tailored for medical image segmentation; the Generalized Probabilistic U-Net (Bhat et al., 2022); and CIMD (Rahman et al., 2023), a recent approach for capturing ambiguity in segmentation.

### 4.4 Evaluation Metrics

To thoroughly evaluate our model on ambiguous segmentation tasks, we use a wide set of metrics. These fall into two categories: standard metrics, which compare a single prediction to a single ground truth, and ambiguity-aware metrics, which measure both the accuracy and diversity of the entire set of predicted segmentations. This combination allows for a complete evaluation capturing how precise individual masks are, as well as how well the model reflects the full range of possible interpretations. **Although we include standard metrics for completeness, our main focus is on ambiguity-aware metrics, as they directly reflect the key goal of our approach: generating diverse and accurate segmentation distributions.** For standard evaluation, we use the Dice Coefficient and Intersection over Union (IoU), which measure geometric overlap. For ambiguity-aware evaluation, we employ a suite of metrics to assess the quality of the generated distribution of samples ($\mathcal{S}$) against the set of ground truth expert annotations ($\mathcal{G}$). These include Combined Sensitivity ($S_c$), Maximum Dice Score ($D_{\max}$), and Diversity Agreement (DA), as proposed by Rahman et al. (2023). A detailed explanation of these metrics is provided in the Technical Appendix. For a comprehensive assessment, we focus on two key metrics in our main analysis:

**Generalized Energy Distance (GED)** Measures the similarity between the distribution of predicted segmentations ($P_X$) and the distribution of ground truth segmentations ($P_Y$).

$$\text{GED}^2(P_X, P_Y) = 2\,\mathbb{E}[d(X, Y)] - \mathbb{E}[d(X, X')] - \mathbb{E}[d(Y, Y')]. \tag{8}$$

where $X, X' \sim P_X$ are independent samples from the model, $Y, Y' \sim P_Y$ are independent ground truth samples, and $d(\cdot, \cdot)$ is a distance metric (e.g., $1 - \text{Dice}$). While foundational, GED can inadequately reward diversity that does not align with the ground truth distribution, motivating the use of the following composite metrics (Rahman et al., 2023).

**Collective Insight (CI) Score** To provide a single, balanced score, we use the CI score, which is the harmonic mean of the three key ambiguity metrics: Combined Sensitivity, Maximum Dice Score, and Diversity Agreement. The harmonic mean ensures that a model must perform well on all three aspects (coverage, accuracy, and diversity) to achieve a high score. This score was proposed in CIMD (Rahman et al., 2023).

$$\text{CI} = \frac{3 \times S_c \times D_{\max} \times \text{DA}}{S_c \cdot D_{\max} + D_{\max} \cdot \text{DA} + S_c \cdot \text{DA}}. \tag{9}$$

## 5 Results and Discussion

### 5.1 Quantitative Results

Our key quantitative finding, presented in **Table 1**, is not only that *AmbiguousTextDiff* achieves state-of-the-art performance, but also that it highlights the sensitivity of ambiguity-aware methods to evaluation protocol and dataset coverage. While CIMD reports strong results on a curated 1585-image subset, its performance deteriorates sharply when sampled and evaluated on the full 3072-image test set, with the CI Score dropping from 0.759 to 0.470. This indicates that its reported performance does not generalize to the full spectrum of clinical ambiguity. In contrast, our model demonstrates robust performance across the complete benchmark, consistently surpassing all baselines. This is particularly evident in the key metrics that capture diversity and distributional accuracy, where the performance gap between our method and the baselines becomes substantially wider. This result showcases the critical role of semantic conditioning in truly capturing clinical

Table 1: Quantitative comparison on the LIDC-IDRI dataset. Our method, AmbiguousTextDiff, significantly outperforms prior work in diversity (CI) and coverage (GED). For CIMD, we report both their published score on their random 1585-image subset and the score on the full 3072-image test set. The performance gap widens considerably under this more comprehensive evaluation. Best results from our method are highlighted in bold. (↓) indicates lower is better, while (↑) indicates higher is better. The lower $D_{max}$ of our method is an expected trade-off for generating more clinically plausible segmentations, as discussed in the text.

| Method | GED ↓ | CI ↑ | $D_{max}$ ↑ |
|---|---|---|---|
| Probabilistic U-Net (Kohl et al., 2018) | 0.353 | 0.731 | 0.892 |
| PHI-Seg (Baumgartner et al., 2019) | 0.270 | 0.736 | 0.904 |
| Generalized Prob. U-Net (Bhat et al., 2022) | 0.299 | 0.707 | 0.905 |
| CIMD (Their 1585 images subset) (Rahman et al., 2023) | 0.321 | 0.759 | 0.915 |
| CIMD (All 3072 images) | 0.306 | 0.470 | 0.684 |
| AmbiguousTextDiff (500 images) | 0.178 | 0.800 | 0.789 |
| **AmbiguousTextDiff (All 3072 images)** | **0.152** | **0.835** | **0.814** |

uncertainty, especially on challenging and ambiguous cases that may have been excluded from smaller, curated test sets.

### 5.1.1 Superior Coverage of Diagnostic Uncertainty

Our most notable finding lies in the model's outstanding performance on the **Generalized Energy Distance (GED)** and **Collective Insight (CI)** metrics. AmbiguousTextDiff achieves a **GED of 0.1523** (lower is better) and a **CI score of 0.8356** (higher is better), both significantly better than the previous state-of-the-art. A low GED indicates that the distribution of our model's predicted segmentations closely matches the range of annotations provided by expert radiologists. At the same time, a high CI score shows that our model generates diverse predictions that collectively capture the true variability found in expert annotations. This improvement is driven by the use of semantic text prompts, which help the model explore a broader and more clinically meaningful range of solutions going beyond predictions based solely on statistical patterns.

### 5.1.2 Interpreting the $D_{max}$ Trade-off

While AmbiguousTextDiff leads in distributional metrics, it reports a slightly lower peak Dice score ($D_{max}$ **of 0.8142**) than models like CIMD (Rahman et al., 2023). Rather than a shortcoming, we view this as a meaningful trade-off and even a strength of our text-conditioned approach. The $D_{max}$ metric favors a single best prediction that most closely matches *one* of the radiologist provided masks. However, our model is guided by semantic prompts, for example, a phrase like "large, spiculated nodule" directs the model to produce segmentations consistent with that specific morphology. It does not generate alternative shapes, such as a "smooth" contour, just to achieve a higher Dice score if such a shape contradicts the given prompt. This semantic grounding encourages the model to prioritize clinical realism over purely numerical optimization. In doing so, AmbiguousTextDiff shifts the goal from simply maximizing overlap with a ground truth mask to accurately capturing the diagnostic ambiguity expressed in natural language. Our strong performance on GED and CI metrics supports this approach, demonstrating that it provides a more complete and clinically meaningful representation of uncertainty in medical image segmentation. Codes and datasets are in supplementary.

## 5.2 Qualitative Analysis

Visual inspection of the segmentations (Figure 4) confirms that *AmbiguousTextDiff* captures a diverse range of interpretations, including variations in boundary smoothness and the inclusion of uncertain regions. To provide a more comprehensive view, we include additional qualitative evidence in the Appendix: Figure 6 showcases the 16-sample diversity, Figure 7 provides a direct visual comparison against the CIMD baseline,

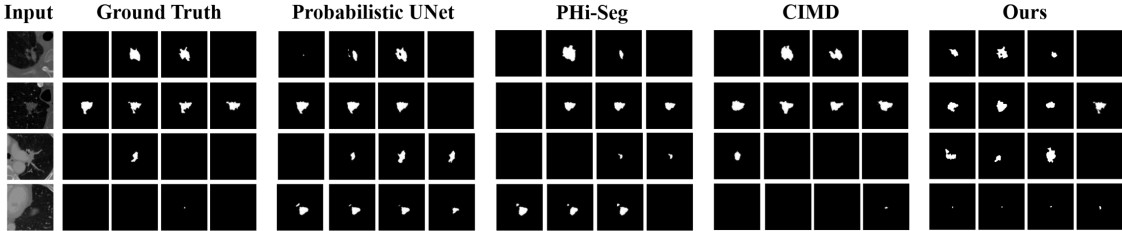

Figure 4: We compare our method with three baselines: Probabilistic U-Net, PHi-Seg, and CIMD using a qualitative analysis. on the left, we show example images from the LIDC-IDRI dataset, each with 4 expert annotations. note that in this dataset, even empty segmentation masks are considered valid expert opinions. to ensure a fair comparison, we display only the first 4 segmentation samples generated by each model. the reader must note that in ambiguous segmentation, we don't match the exact segmentation with ground truth, but look at variability in output as captured by a suitable metric such as GED. for example, in row three, we see the ground truth has many empty segmentations, but some methods including ours try to capture possible variability.

and Figure 9 displays results for the IMA++ dataset. Furthermore, the model's response to contradictory semantic prompts is visualized in the semantic control experiment in Figure 8.

### 5.2.1 Cross-Modality Validation on IMA++

To evaluate the generalizability of AmbiguousTextDiff, we conducted experiments on the IMA++ skin lesion dataset. As shown in **Table 2**, our method significantly outperforms probabilistic baselines. Specifically, our model achieves a **GED of 0.0724**, which is nearly double the performance of the CIMD baseline (0.1389). This indicates that the text-guided framework is highly effective at resolving ambiguities caused by low-contrast lesion boundaries and common dermoscopic artifacts like hair or gel bubbles.

Table 2: Quantitative comparison on the IMA++ (Skin Dermoscopy) dataset. This cross-modality validation demonstrates that AmbiguousTextDiff generalizes effectively to color-based imaging and handles distinct ambiguity characteristics like low-contrast lesion boundaries and artifacts. Best results are in bold.

| Method | GED $\downarrow$ | CI $\uparrow$ | $D_{max}$ $\uparrow$ | $S_c$ $\uparrow$ | DA $\uparrow$ |
|---|---|---|---|---|---|
| Probabilistic U-Net (Kohl et al., 2018) | 0.2923 | 0.7713 | 0.8498 | 0.8488 | 0.9283 |
| PHI-Seg (Baumgartner et al., 2019) | 0.1950 | 0.7830 | 0.8405 | 0.9056 | 0.8966 |
| Generalized Prob. U-Net (Bhat et al., 2022) | 0.3371 | 0.7067 | 0.8269 | 0.7782 | 0.9033 |
| CIMD (Rahman et al., 2023) | 0.1389 | 0.7955 | 0.8988 | 0.9254 | 0.8578 |
| **AmbiguousTextDiff (Ours)** | **0.0724** | **0.8848** | **0.9510** | **0.9974** | **0.7538** |

## 6 Conclusion

We introduced *AmbiguousTextDiff*, a novel text-guided diffusion model designed to address the fundamental challenge of ambiguity in medical image segmentation. By conditioning the generative process on rich, semantically meaningful natural language prompts, our model generates diverse yet clinically consistent segmentation proposals rather than a single "ground truth" mask, thereby modeling the inherent uncertainty present in complex medical images. By validating our framework on two distinct imaging modalities, thoracic CT (LIDC-IDRI) and skin dermoscopy (IMA++), we demonstrate that semantic conditioning robustly captures inter-observer variability across different anatomical domains. Our results prove that text-guided models for ambiguous segmentation are not only more accurate but also aware of clinically relevant ambiguity. This multi-modal validation highlights the generalizability of our approach and its potential to function as an interpretable reasoning tool in real-world clinical settings.

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

# A  Appendix

## A.1  Implementation and Experimental Details

**Dataset Splits and Preprocessing.**  We follow a subject-level split to prevent data leakage between training and test sets. From the 1012 subjects in the LIDC-IDRI dataset, 80% were used for training and 20% for testing. All grayscale CT slices were resized to $128 \times 128$ pixels and normalized to the range $[-1, 1]$. Slices with no annotations from any of the four radiologists were treated as having empty masks.

**Diffusion Model.**  We used a diffusion process with $T = 1000$ timesteps and a linear noise schedule for $\beta_t$ defined as:

$$\beta_t = \beta_1 + \frac{t-1}{T-1}(\beta_T - \beta_1), \tag{10}$$

where $\beta_1 = 10^{-4}$ and $\beta_T = 0.02$. The denoising backbone is a U-Net with a base channel size of 128. Both self-attention and cross-attention modules were implemented using multi-head attention, with 64 channels per head. The model was trained using the AdamW optimizer with an initial learning rate of $1 \times 10^{-4}$, decayed linearly over 50,000 steps. Training was run for 50,000 steps in total, and checkpoints were saved every 20,000 steps. Gradient clipping (max norm = 1.0) were applied for regularization and stability.

The model was trained on a composite loss function combining the standard diffusion reconstruction loss ($\mathcal{L}_{\mathrm{MSE}}$) in pixel space with a KL divergence term ($\mathcal{L}_{\mathrm{KL}}$) that regularizes the model's predictions in a learned latent space:

$$\mathcal{L}_{\mathrm{total}} = \mathcal{L}_{\mathrm{MSE}} + \lambda \mathcal{L}_{\mathrm{KL}}. \tag{11}$$

We set the weighting factor $\lambda$ to **0.001**. This value was determined empirically to balance the numerical scales of the two loss components. It ensures that the primary pixel-level denoising objective is not overpowered by the latent space regularizer, allowing the KL term to guide the model toward contextually appropriate segmentations while maintaining a stable training process.

**Text Encoder.**  To incorporate textual guidance, we used the `Bio_ClinicalBERT` model (Alsentzer et al., 2019) as a frozen text encoder. Prompts derived from LIDC-IDRI metadata were tokenized and passed through the BERT model to obtain 768-dimensional embeddings. These embeddings were then linearly projected to match the 128-dimensional channel size used in the U-Net's attention layers.

**EMA and Optimization Techniques.**  We employed Exponential Moving Average (EMA) to maintain a smoothed version of model weights during training, following the update rule:

$$\theta_{\mathrm{EMA}} = \beta \, \theta_{\mathrm{EMA}} + (1 - \beta) \, \theta_{\mathrm{current}}, \quad \beta = 0.9999.$$

This technique helps stabilize training and improves generalization.

**Training Details.** Our framework was implemented in PyTorch and trained on 4 NVIDIA RTX 4090 GPUs for 50,000 steps with a batch size of 4. We used the AdamW optimizer with a learning rate of $1 \times 10^{-4}$ (linearly annealed to zero) and no weight decay. An Exponential Moving Average (EMA) with a decay rate of 0.9999 was maintained, and gradients were clipped at a max norm of 1.0. The KL-divergence loss weight, $\lambda$, was set to 0.001.

**Inference Details.** During inference, we generated $N = 16$ samples per test image using the standard DDPM sampler with $T = 1000$ steps, without classifier-free guidance or conditioning dropout. Model outputs were continuous masks in the range $[-1, 1]$, binarized at 0.5 to produce segmentation masks for evaluation. The 500-image subset used for initial validation was randomly sampled from the full test set.

**Ablation Study** All ablation experiments, including "w/o Text Guidance" and "w/o Ambiguity Modeling," were conducted using the same training and evaluation as our main model, trained on the full training set and evaluated on the full test set to ensure a fair and direct comparison. The results in Table 5 are all reported in the full test set, with the "Ours (500 images)" row included for reference to our initial pilot study.

## A.2 Detailed Evaluation Metrics

### A.2.1 Standard Metrics

These metrics are used to measure the geometric similarity between a single predicted mask, $X$, and a single ground truth annotation, $Y$.

While they are commonly used in deterministic segmentation tasks, these metrics have clear limitations in our setting. Since they compare only one prediction to one ground truth, they cannot fully capture how well a model represents the range of possible valid segmentations. A model might still achieve a high score on these metrics while failing to reflect the true ambiguity present in the data. Nonetheless, we include them for a complete and fair comparison with existing methods.

**Dice Coefficient (Dice)** Measures the overlap between two binary masks. It is sensitive to the size of the segmented region and is one of the most common metrics in medical image segmentation.

$$\text{Dice}(X, Y) = \frac{2|X \cap Y|}{|X| + |Y|}.$$ (12)

**Intersection over Union (IoU)** Also known as the Jaccard Index, this metric quantifies the ratio of the intersection to the union of the predicted and ground truth masks.

$$\text{IoU}(X, Y) = \frac{|X \cap Y|}{|X \cup Y|}.$$ (13)

### A.2.2 Ambiguity-Specific Metrics

Standard metrics are insufficient for ambiguous segmentation, as they do not account for the multiple valid ways an object can be segmented. The following metrics assess the quality of a generated distribution of samples. Let $\mathcal{S} = \{s_1, s_2, \ldots, s_n\}$ be the set of $n$ generated sample masks from our model, and $\mathcal{G} = \{g_1, g_2, \ldots, g_m\}$ be the set of $m$ ground truth masks from multiple annotators.

**Combined Sensitivity ($S_c$).** This metric evaluates the collective coverage of the generated sample ensemble ($\mathcal{S}$) against the complete set of ground truth annotations ($\mathcal{G}$). It computes the sensitivity (recall) of the union of all generated masks with respect to the union of all ground truth masks. First, we define the union masks:

$$S_U = \bigcup_{i=1}^{n} s_i \quad \text{and} \quad G_U = \bigcup_{j=1}^{m} g_j.$$ (14)

The Combined Sensitivity is then given by:

$$S_c = \frac{|S_U \cap G_U|}{|G_U|}. \tag{15}$$

A higher $S_c$ indicates that the model's predictions collectively capture all regions marked by any expert.

**Maximum Dice Score ($D_{\mathbf{max}}$).** This metric assesses how well each ground truth mask in $\mathcal{G}$ is represented by at least one of the generated samples in $\mathcal{S}$. It finds the best-matching generated sample for each ground truth mask and averages these maximum Dice scores.

$$D_{\max} = \frac{1}{|\mathcal{G}|} \sum_{g \in \mathcal{G}} \max_{s \in \mathcal{S}} \mathrm{Dice}(s, g). \tag{16}$$

**Diversity Agreement (DA).** Quantifies how well the diversity in the model's generated samples aligns with the diversity observed in the human-annotated ground truths. It penalizes both under- and over-diverse predictions.

$$\mathrm{DA} = 1 - \frac{|\Delta V_{\min}| + |\Delta V_{\max}|}{2}. \tag{17}$$

where:

- Let $V_{\min}^{\mathrm{GT}}, V_{\max}^{\mathrm{GT}}$ be the minimum and maximum dissimilarities (e.g., $1 - \mathrm{Dice}$) among pairs of ground truth segmentations.
- Let $V_{\min}^{S}, V_{\max}^{S}$ be the minimum and maximum dissimilarities among pairs of generated segmentations. Then we define
- $\Delta V_{\min} = V_{\min}^{\mathrm{GT}} - V_{\min}^{S}$ and $\Delta V_{\max} = V_{\max}^{\mathrm{GT}} - V_{\max}^{S}$.

A DA score close to 1 indicates a perfect match in the range of diversity.

### A.3 Inference Speed and Hardware Adaptability

While our primary evaluation uses a 1000-step DDPM sampler for maximum fidelity, we evaluated the model using a DDIM sampler with 100 steps to assess hardware adaptability. As shown in Table 3, DDIM provides a 3x speedup with only a marginal impact on distributional metrics.

Table 3: Inference speed comparison ($N = 16$ samples). Evaluation performed on 4 NVIDIA RTX 4090 GPUs.

| Method | Steps | GED $\downarrow$ | CI $\uparrow$ | Total Inf. Time |
|---|---|---|---|---|
| AmbiguousTextDiff (DDPM) | 1000 | 0.152 | 0.835 | $\sim$69h |
| AmbiguousTextDiff (DDIM) | 100 | 0.177 | 0.809 | $\sim$23h |

### A.4 Standard Deterministic Metrics on LIDC-IDRI

Table 4 provides these for the full LIDC-IDRI set. Note that these metrics are lower than typical segmentation tasks because the dataset includes expert disagreements where some experts provide empty masks (no nodule detected), which heavily penalizes single-point estimates.

### A.5 Ablation Study

To better understand the design of *AmbiguousTextDiff* and quantify the importance of its core components, we performed a series of ablation studies. As shown in Table 5 (and with qualitative results in Figure 5 in the appendix), the results clearly demonstrate that both text guidance and the ambiguity modeling objective play a crucial role in achieving the model's state-of-the-art performance. Removing either leads to a noticeable drop in effectiveness, with the model producing less diverse or less clinically relevant segmentations. A detailed breakdown of each ablation experiment is provided in the Appendix.

Table 4: Standard Mean Dice and IoU on LIDC-IDRI.

| Method | Avg. Dice ↑ | Avg. IoU ↑ |
|---|---|---|
| CIMD | 0.298 | 0.277 |
| **AmbiguousTextDiff (Ours)** | **0.392** | **0.340** |

Table 5: Ablation study of AmbiguousTextDiff components. The ablated models show a clear performance drop, underscoring the importance of both text guidance and the ambiguity modeling objective. Here, 500 and All denote the number of images, Ours refers to our method AmbiguousTextDiff, while DA and $S_c$ represent Diversity Agreement and Combined Sensitivity, respectively.

| Method | CI↑ | $S_c$ ↑ | $D_{max}$ ↑ | DA ↑ | GED ↓ |
|---|---|---|---|---|---|
| Ours (500) | 0.800 | 0.875 | 0.789 | 0.865 | 0.178 |
| **Ours (All)** | **0.835** | **0.895** | **0.814** | **0.885** | **0.152** |
| w/o Text Guidance | 0.766 | 0.876 | 0.743 | 0.864 | 0.214 |
| w/o Ambiguity Modeling | 0.716 | 0.773 | 0.731 | 0.851 | 0.173 |

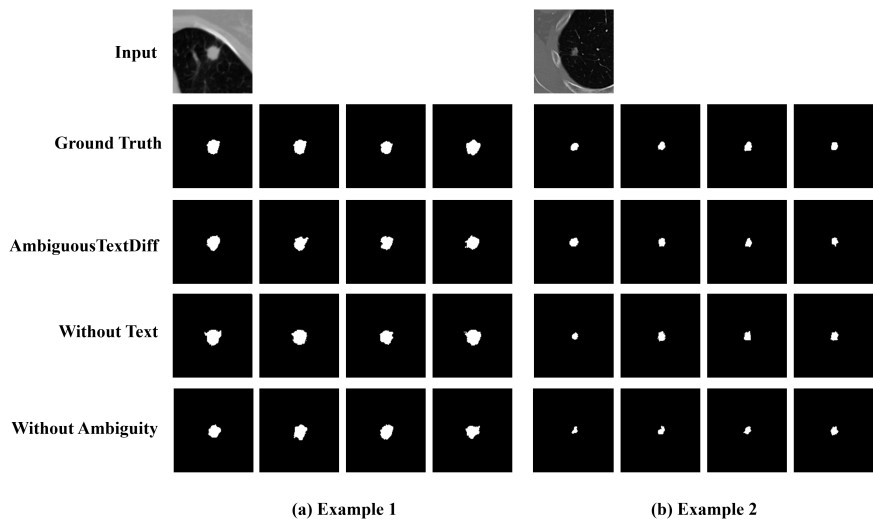

(a) Example 1          (b) Example 2

Figure 5: **Ablation analysis on the LIDC-IDRI dataset comparing *AmbiguousTextDiff* against its variants.** each column shows a separate example. the topmost image is the input CT scan slice, followed by segmentation masks from four expert annotators (*ground truth*). we compare: (i) **AmbiguousTextDiff** (our full model), (ii) **without text** (removing textual guidance), and (iii) **without ambiguity** (removing ambiguity modeling). for fairness, the first 4 sampled masks from each model are visualized. note that empty masks are valid annotations in LIDC-IDRI. *AmbiguousTextDiff* better reflects expert-level variability, while removing text or ambiguity leads to overconfident or less diverse outputs.

## A.6 Detailed Ablation Analysis

Here we provide a more detailed breakdown of the ablation studies summarized in the main paper.

### A.6.1 The Critical Role of Text Guidance

**Methodology** To understand the impact of semantic conditioning, we conducted an ablation where text-based guidance was removed. In this variant, referred to as "w/o Text Guidance," we skipped the text encoder and instead used features from the input image extracted using the same image encoder already present in the model as the conditioning signal. This allowed the model to rely solely on visual information, without

any explicit semantic prompts, enabling a clearer assessment of how much the natural language guidance contributes to segmentation performance.

To better understand the design of *AmbiguousTextDiff* and quantify the importance of its core components, we performed a series of ablation studies. Specifically, we systematically removed two key elements: text guidance and the ambiguity modeling objective, and evaluated the resulting performance on the full LIDC-IDRI dataset.

As shown in **Table 5**, the results clearly demonstrate that both components play a crucial role in achieving the model's state-of-the-art performance. Removing either leads to a noticeable drop in effectiveness, highlighting their complementary contributions to modeling segmentation uncertainty. The qualitative results in **Figure 5** further illustrate these performance differences.

**Analysis**  As shown in **Table 5**, removing text guidance results in a noticeable drop in performance. The most prominent decline is observed in $D_{\mathbf{max}}$, which decreases from 0.814 to 0.743, while **GED** increases from 0.152 to 0.214, indicating poorer alignment with the distribution of ground truth annotations. These results suggest that without the semantic grounding provided by text prompts, the model struggles to generate segmentations that are both accurate (as reflected in Dice) and representative of the variability seen in expert annotations (captured by GED). Although the model still produces some variation, the outputs tend to be less clinically meaningful and often deviate from any realistic interpretation. This experiment highlights that text guidance is not just an auxiliary feature, it plays a central role in guiding the model towards generating semantically rich and diagnostically relevant segmentations.

### A.6.2  The Necessity of Explicit Ambiguity Modeling

**Methodology.**  We next explored the role of our explicit ambiguity modeling objective by ablating the KL-divergence term from the loss function. The resulting variant, labeled "w/o Ambiguity Modeling," effectively disables the model's incentive to learn the underlying distribution of expert annotations. Instead, it learns to denoise a single, randomly selected ground-truth mask at each step, behaving like a conventional text-guided diffusion model (Baranchuk et al., 2021; Feng, 2024) with no awareness of variability across annotators.

**Analysis.**  This ablation underscores the crucial role of our probabilistic training objective. When the KL regularization is removed, the **CI Score** drops sharply from 0.8356 to 0.7160, indicating that the model fails to capture the full range of expert-provided annotations. We also observe a decline in Diversity Agreement (**DA**), pointing to mode collapse; where the model produces fewer distinct segmentations and gravitates toward a narrow, oversimplified prediction space. Further evidence comes from worsened scores on **GED**, **Combined Sensitivity**, and $D_{\mathbf{max}}$, all of which reinforce the conclusion that without KL regularization, the model loses its ability to represent diagnostic uncertainty. These findings confirm that the KL objective is not just a regularization term; it is essential for encouraging diverse, clinically meaningful outputs that reflect the true variability in expert interpretations.

### A.6.3  Traditional Segmentation Metrics

Traditional metrics such as Dice and Intersection-over-Union (IoU), while commonly used in deterministic segmentation tasks, are limited in their ability to evaluate models that generate diverse output distributions. These metrics tend to favor predictions that resemble an "average" of the ground truth masks, often penalizing outputs that are diverse yet clinically meaningful; precisely the type of predictions our model is designed to produce.

Nonetheless, we report mean Dice and IoU scores across all model variants for completeness. On the full LIDC-IDRI dataset, AmbiguousTextDiff achieved Dice and IoU scores of 0.3925 and 0.3406 respectively. On the 500-image pilot subset, the scores were 0.3845 and 0.3328 respectively. The variant without text guidance obtained 0.3801 and 0.3315, while the model without ambiguity modeling scored 0.3871 and 0.3388 respectively.

These relatively small differences despite significant changes in distributional quality highlight the limitations of traditional metrics. They fail to capture the nuanced improvements in diversity, uncertainty modeling, and clinical relevance delivered by our approach. This reinforces the importance of using distribution-aware metrics, as reported in Table 5, for evaluating models like AmbiguousTextDiff.

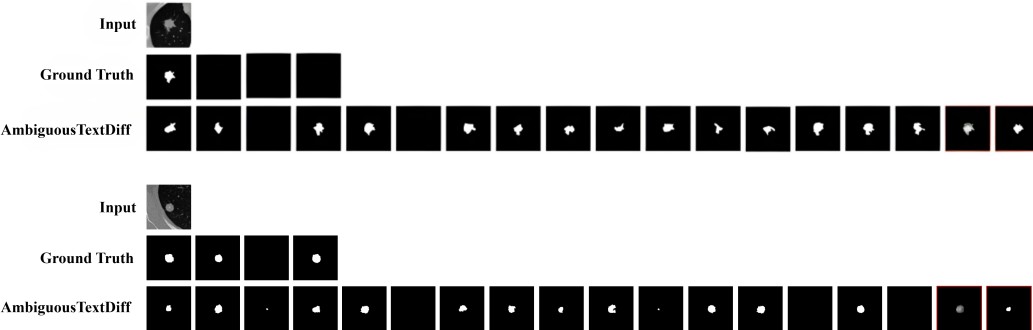

Figure 6: Few examples of segmentation using the AmbiguousTextDiff model. The figure displays the input lung nodule from a CT scan and the corresponding ground truth. The bottom row presents 16 unique samples generated by the model, followed by the consolidated ensemble average and the final majority vote segmentation.

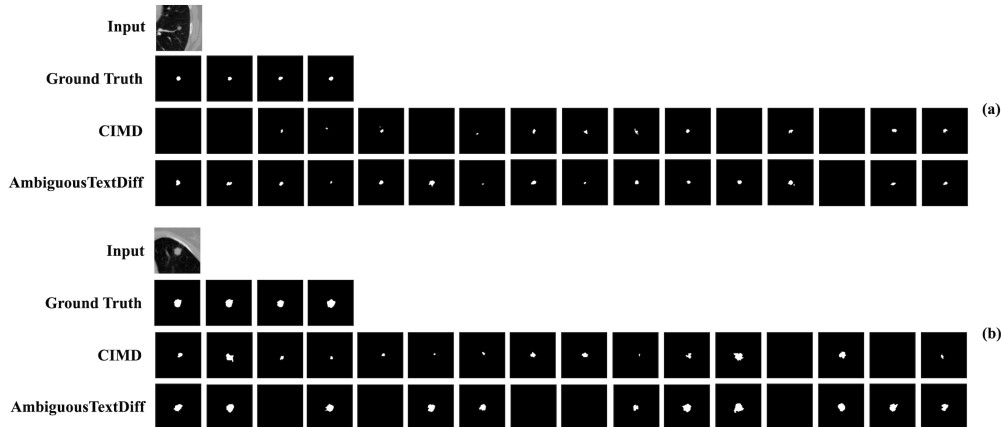

Figure 7: Qualitative comparison of lung nodule segmentation. For each input CT slice, four expert-provided ground truth annotations are shown. Results from CIMD were reproduced using the official GitHub repository from their paper, while AmbiguousTextDiff shows 16 diverse segmentation samples generated by our model. Both methods were trained on the full test set of 3,072 images. (a) and (b) depict two representative examples.

## A.7 Choice of Pixel-Space Diffusion.

Our model performs the diffusion process directly in the pixel space rather than a compressed latent space, as seen in Latent Diffusion Models (LDMs) (Rombach et al., 2022). While LDMs offer significant computational efficiency, we opted for a pixel-space model for two primary reasons. First, medical image segmentation requires extremely high fidelity and preservation of fine-grained details, such as subtle texture and irregular boundaries, which can sometimes be lost or distorted during the compression and decompression stages of an LDM's autoencoder. Second, by operating directly on the pixels, our model avoids introducing an additional source of architectural complexity and potential information loss, allowing for a more direct study of the effects of text guidance and ambiguity modeling. We acknowledge the computational cost as a limitation and view the integration of these principles into an efficient LDM framework as a promising direction for future work.

### A.8 Qualitative Validation of Semantic Control

While our quantitative results demonstrate the model's overall performance, a crucial aspect of our work is its ability to interpret and act upon the semantic content of text prompts. To better validate this capability, we carried out a focused qualitative experiment to clearly test the impact of textual guidance. We chose two challenging test cases with notable expert disagreement in their ground truth annotations and conditioned our model on contradictory descriptive prompts for the same input image. The complete results are presented in Figure 8.

**Example 1: Control over Nodule Texture ('Spiculated' vs. 'Smooth')**

The first case, the Figure 8 features a highly subtle nodule where expert annotations varied greatly, with one radiologist providing an empty mask. The textual metadata from the dataset for this nodule is particularly revealing; for instance, the full prompt from one of the annotation includes the descriptions: 'The Subtlety is Fairly Subtle. The Sphericity is Ovoid/Round. The Spiculation is No Spiculation.'. Such descriptions explicitly capture the uncertainty in how the nodule should be interpreted.

We first conditioned the model with a prompt aligned to one possible interpretation: "A lung nodule with marked spiculation." (Prompt A). The generated samples reflect the inherent ambiguity, spanning from an entirely empty mask (matching the dissenting expert) to several small segmentations with jagged, irregular contours consistent with spiculation.

Next, using the same input image, we provided a semantically contradictory prompt: "A lung nodule that is smooth and round." (Prompt B). The model responded clearly, producing segmentations with more compact shapes and smoother, rounded boundaries; directly adapting its output to the new textual guidance.

**Example 2: Control over Nodule Margin ('Irregular' vs. 'Round')**

To test the consistency of this effect, the second case, the Figure 8 presents a visually complex and irregular region where three of four experts provided empty masks. Here, the textual metadata strongly aligns with the visual evidence. The full prompt for one of the annotation includes key descriptors such as: 'The Margin is Near Poorly Defined. The Lobulation is Marked Lobulation. The Spiculation is Marked Spiculation.'.

When guided by a factual prompt summarizing this description: "An irregular lung nodule" (Prompt A): the model's outputs reflected expert consensus. Most samples were empty, and the few non-empty masks were highly fragmented and irregular, consistent with the "irregular" characterization.

The stronger test came with the contradictory prompt: "A perfectly round lung nodule" (Prompt B). Despite the visually complex evidence, the model adapted to the textual instruction, generating masks that were noticeably simpler and more rounded than those from Prompt A. This demonstrates the model's ability to enforce semantic constraints from text even when they conflict with the underlying image data.

Overall, these experiments demonstrate strong evidence of genuine semantic control. The model not only interprets high-level concepts (e.g., "spiculated," "round") but also translates them into the corresponding geometric characteristics of its segmentations. By grounding the evaluation in the dataset's full descriptive prompts, we show that the model can generate outputs aligned with factual descriptions and, more importantly, flexibly adjust when given contradictory semantic instructions. This highlights the central contribution of our work: a controllable and ambiguity-aware segmentation model.

### A.9 Analysis of the Number of Generated Samples

An important aspect of our experimental setup is the number of samples ($N$) generated per input image. To justify our choice and investigate the trade-off between diversity and computational cost, we performed an ablation study by varying $N$ from 1 to 32. The goal was to identify when additional samples stop contributing meaningfully to capturing the ground-truth ambiguity.

The results, illustrated in Figure 10, reveal a clear trend. For lower values of $N$ (1 to 8), performance improves significantly highlighting that a single deterministic output is insufficient for modeling diagnostic uncertainty. As $N$ increases to 16, most key metrics: **CI Score**, **Combined Sensitivity**, $D_{\mathbf{max}}$, and

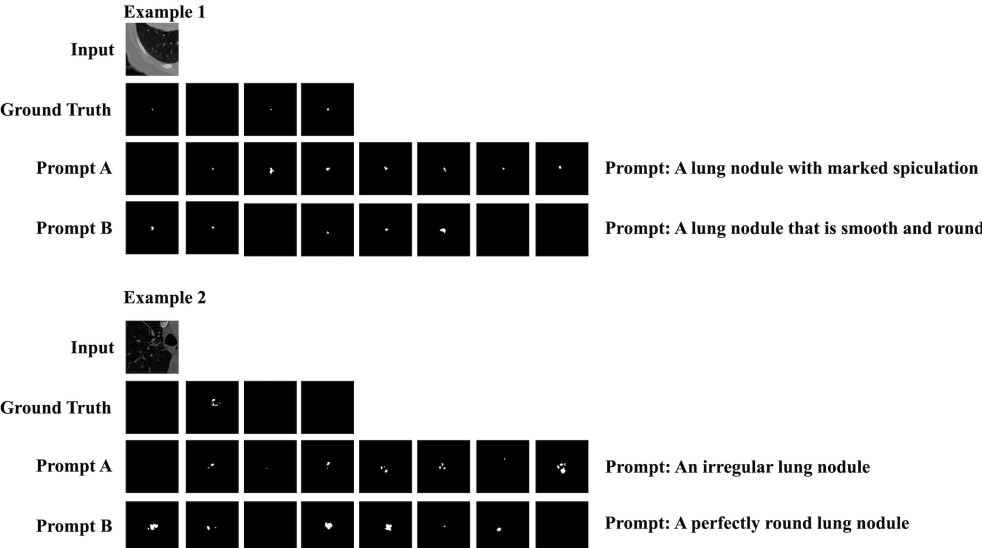

Figure 8: Qualitative validation of semantic control across two distinct examples. For each input image (top row of each example), we generate 8 segmentation samples using both a factually descriptive prompt (Prompt A) and a semantically contradictory prompt (Prompt B), with expert-annotated ground truths provided for reference. The model's outputs consistently adapt to the textual description, producing shapes with the requested geometric properties (e.g., spiculated vs. smooth in Example 1; irregular vs. round in Example 2), demonstrating its ability to interpret and act on semantic guidance while respecting underlying image ambiguity.

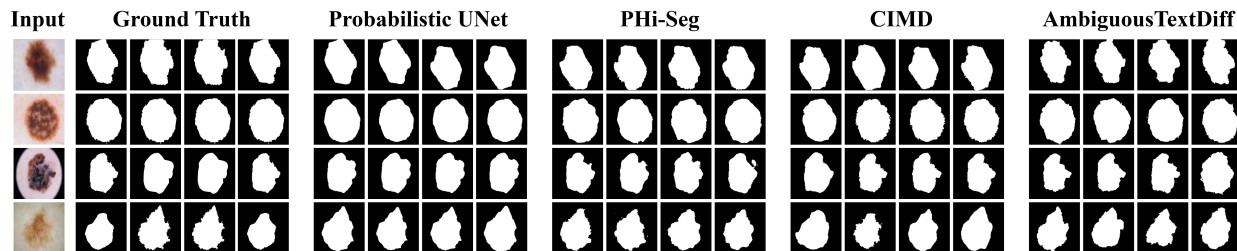

Figure 9: We compare our method with three baselines: Probabilistic U-Net, PHi-Seg, and CIMD using a qualitative analysis. on the left, we show example images from the IMA++ dataset, each with 4 expert annotations. To ensure a fair comparison, we display only the first 4 segmentation samples generated by each model. the reader must note that in ambiguous segmentation, we don't match the exact segmentation with ground truth, but look at variability in output as captured by a suitable metric such as GED.

**Diversity Agreement** either peak or approach saturation, indicating comprehensive coverage of the expert annotation space.

Beyond $N = 16$, however, we observe a slight decline in performance. Metrics such as the **CI Score** and $D_{\mathbf{max}}$ decrease, while **GED** increases indicating a weaker alignment with the target distribution. A closer qualitative inspection reveals that many of the additional samples beyond this point are blank (i.e., entirely black masks), offering no meaningful contribution to the ensemble. These empty or degenerate outputs reduce the overall diversity and coverage of the predictions, effectively introducing noise into the distribution.

This analysis confirms that $\mathbf{N = 16}$ strikes an optimal balance: it ensures high-quality, diverse predictions without introducing unnecessary redundancy or invalid samples. This choice not only improves efficiency but also enhances the clinical relevance and reliability of the model's outputs. As such, $N = 16$ is used throughout all main experiments, consistent with findings from prior probabilistic segmentation work (Rahman et al., 2023). We show few examples of 16 generated samples for our method in Figures 6 and 6.

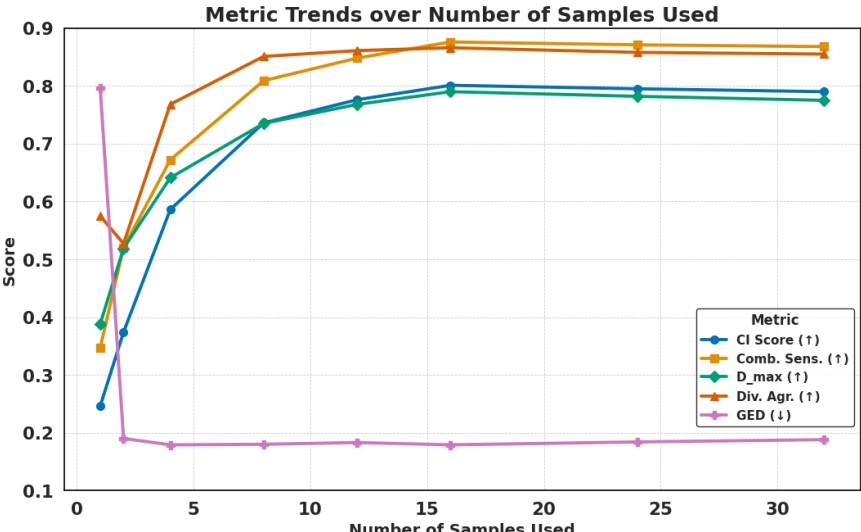

Figure 10: Trends in evaluation metrics as the number of generated samples ($N$) increases. Most metrics improve noticeably up to $N = 16$, after which performance either levels off or begins to decline slightly. This suggests that $N = 16$ offers an effective balance-capturing the breadth of diagnostic ambiguity while avoiding the inclusion of redundant or low-quality samples.

