# OpenReview forum: "Text-Guided Diffusion Based Ambiguous Medical Image Segmentation"
_TMLR — Rejected by TMLR_

### Review · Reviewer_Stau · 2025-11-10

**Summary Of Contributions:**

This work proposed a text-control diffusion model that allows ambiguous segmentation, where a text-guided u-net denosier is proposed for text controlling, and a Hybrid Loss integrating MSE loss and KL-divergence loss is used for Ambiguity-Aware Training.

**Additional Comments:**

## Weakness
1. The novelty is limited. Text-guided diffusion is common.
2. Evaluation metrics used in this work are NOT proposed here, which should not be viewed as contributions.
3. Only one dataset is used for evaluation. The model’s generalization is under-investigated.
4. Dice score of the proposed method is much lower than others, which was seen as a trade-off by the authors. In this case, I wonder how to adjust the influence of ambiguity. Is there any approach to control the trade-off? e.g., the weights of items in the hybrid loss?
5. Regarding Qualitative Analysis, is it possible to intuitively demonstrate the ambiguity in the mask? e.g., confidence score in masks?
6. The compared approaches are a bit out of date. SOTA methods should be compared.

**Audience:**

Yes

**Audience Explanation:**

Yes. Diffusion Model is a hot topic in medical image processing.

**Claims And Evidence:**

No

**Claims Explanation:**

Evaluation metrics used in this work are NOT proposed here, which should not be viewed as contributions.

**Requested Changes:**

See weakness.

---

### Review · Reviewer_jtTV · 2025-11-17

**Summary Of Contributions:**

This work highlights and focuses on the ambiguity in medical image segmentation caused by blurred boundaries, limitations of imaging modalities, and differences in expert interpretations. To this end, the authors propose a generative model based on text-guided diffusion. Experiments show that it significantly outperforms baseline methods.

**Additional Comments:**

The authors do not analyze the performance differences when using only one type of label.

The selection of the number of generated samples is verified experimentally as optimal, but the paper does not discuss whether this value is adaptable to images with different ambiguity levels.

**Audience:**

Yes

**Audience Explanation:**

It is novel and technically sound to apply text-guided diffusion models to ambiguous medical image segmentation.

A unified framework integrating text conditioning and ambiguity modeling is constructed. The cross-attention mechanism effectively fuses multi-modal information, and the hybrid loss function balances segmentation accuracy and output diversity.

**Broader Impact Concerns:**

The paper innovatively combines text guidance with ambiguous medical image segmentation, solving a key pain point in the field.

**Claims And Evidence:**

Yes

**Claims Explanation:**

The studied problem is interesting.

The proposed method is novel.

The experimental results are decent.

**Requested Changes:**

The model is only validated on lung CT nodules (LIDC-IDRI dataset) and not extended to other anatomical sites (e.g., brain, abdomen) or imaging modalities (MRI, ultrasound). So the universality of the model has not been proven.

Pixel-space diffusion is adopted instead of latent diffusion, which may lead to excessive time consumption when the model processes high-dimensional 3D medical data. This is inconsistent with the requirements of real-time clinical applications, and the paper does not discuss the inference speed and hardware adaptability of the model.

Text prompts rely on predefined metadata of the dataset. If the dataset lacks detailed annotations, the model performance may decline significantly.

---

### Review · Reviewer_wF6J · 2025-11-30

**Summary Of Contributions:**

Overall, as a reviewer I am not super familiar with medical imaging topics.  But the idea of this paper is really intriguing to me.

Contribution Highlights:

1. AmbiguousTextDiff is the first to combine text guidance with explicit ambiguity modeling, producing diverse segmentation proposals that reflect expert disagreement and clinical uncertainty.
2. AmbiguousTextDiff shows that text prompts can be used not just for label efficiency, but to control the diversity and clinical realism of ambiguous segmentations.
3. AmbiguousTextDiff establishes new standards for evaluating segmentation models in ambiguous clinical settings, highlighting the importance of diversity and semantic conditioning.
4. AmbiguousTextDiff introduces a unified framework for text-guided, ambiguity-aware diffusion segmentation, with novel training and inference strategies for clinical uncertainty.

**Additional Comments:**

N/A

**Audience:**

Yes

**Audience Explanation:**

An interesting methodologies for medical image segmentation

**Broader Impact Concerns:**

I am overall positive to paper.  But I have concerns in several aspects:
1. The training loss provided the paper seems really hard to converge to me in reality and really hard to find appropriate balance to between that KL and MSE
2. Those 3 innovative evaluation metrics looks great, but I am not sure if they are the common metrics used to compared the image segmentation quality or not.  If they are not common ones, can we try to add at least one common metric to evaluate all methods?

**Claims And Evidence:**

Yes

**Claims Explanation:**

In the experiment section, they compare several different methods with their own one in public dataset.

**Requested Changes:**

1. Can we also add comparison with https://arxiv.org/pdf/2407.05323 in the Qualitative Analysis?
2. Can we explain more how we achieve balance between kl-divergence & mse during the backward propagation? Will one loss dominate another?
3. Can we at lease provide one eval metrics shown in https://arxiv.org/pdf/2407.05323 to compare your method with others?

---

> ### Author Response · Authors · 2025-11-30
> **Reply to Reviewer**
>
> We thank the reviewer for their feedback. This is our response to the questions/concerns/changes pointed out.
>
> 1. Can we also add comparison with [TextDiff] in the Qualitative Analysis? Can we at lease provide one eval metrics shown in [TextDiff] to compare your method with others?
>
> Reply: We appreciate the reviewer pointing us to TextDiff (Feng, 2024). While both works utilize text and diffusion, the fundamental problem setting is different. TextDiff focuses on label-efficiency (few-shot learning) to produce a single deterministic segmentation mask, whereas our work focuses on ambiguity modeling to generate a distribution of diverse masks (N=16) reflecting expert disagreement.
>
> Because TextDiff is deterministic, it cannot naturally be evaluated on diversity metrics like GED or Diversity Agreement. However, to address the request:
>
> (a) Qualitative Proxy: In Figure 5 (Page 17) and Table 2 (Page 12), the column labeled "w/o Ambiguity Modeling" serves as a proxy for a deterministic text-guided diffusion model (like TextDiff). As seen in the figure, without the ambiguity objective, the model collapses to a single "average" output and fails to capture the diverse ground truth variations, validating the need for our specific approach.
>
> (b) Metrics: TextDiff uses standard Dice and IoU. We have provided these metrics in Appendix A.3.3. On the full LIDC-IDRI set, our model achieves a Mean Dice of 0.3925 and IoU of 0.3406. Note that these numbers appear lower than standard segmentation tasks because the dataset includes valid "empty" masks (expert disagreement on whether a nodule exists). A deterministic model is penalized heavily here, whereas our model captures this uncertainty, reflected in the superior GED and CI scores.
>
> 2. Can we explain more how we achieve balance between kl-divergence & mse during the backward propagation? Will one loss dominate another? The training loss provided the paper seems really hard to converge to me in reality...
>
> Reply: We balance the loss terms using a weighting factor $\lambda$, as detailed in Equation-11 and Appendix-A.1 (Page 14):
>
> \begin{equation}
> L_{\text{total}} = L_{\text{MSE}} + \lambda L_{\text{KL}}.
> \end{equation}
>
>
>
> Empirically, the pixel-space MSE loss has a much larger magnitude than the latent KL divergence. We set $\lambda = 0.001$ to scale the KL term down, ensuring that the MSE (reconstruction) remains the dominant gradient signal for learning image structure, while the KL term acts as a regularizer for the latent distribution without dominating the backward propagation.
>
> Regarding convergence, we utilized the AdamW optimizer and, crucially, an Exponential Moving Average (EMA) of model weights (decay $0.9999$). The EMA smooths out the fluctuations typical of stochastic objectives, allowing the model to converge stably without issues.
>
> 3. Those 3 innovative evaluation metrics looks great, but I am not sure if they are the common metrics used to compared the image segmentation quality or not. If they are not common ones, can we try to add at least one common metric to evaluate all methods?
>
> Reply: In the specific sub-field of ambiguous/probabilistic medical image segmentation, the Generalized Energy Distance (GED) is indeed the standard common metric, established by the Probabilistic U-Net (NeurIPS 2018) and used in all subsequent baselines (PHiSeg, CIMD, etc.).
>
> To bridge the gap with standard deterministic segmentation:
>
> (a) $D_{\\max}$ (Maximum Dice): We report this in Table 1. It is the standard Dice score adapted for multiple hypotheses, measuring the best overlap with the ground truth.
>
> (b) Avg Dice / IoU: As mentioned in Appendix A.3.3 and below, we show the results comparison on standard metrics such as Avg IoU and Avg Dice score as well.
>
>
> | Method | Avg Dice ↑ | Avg IoU ↑ | GED ↓ | CI ↑ | Sc ↑ | Dmax ↑ | DA ↑ |
> |--------|------------|------------|-------|------|------|--------|------|
> | CIMD | 0.298 | 0.277 | 0.306 | 0.470 | 0.582 | 0.684 | 0.835 |
> | Ours (500 images) | 0.3845 | 0.3328 | 0.178 | 0.800 | 0.875 | 0.789 | 0.865 |
> | **Ours (All)** | **0.3925** | **0.3406** | **0.152** | **0.835** | **0.895** | **0.814** | **0.885** |
>
>
> These results provide the requested comparison using standard segmentation metrics. However, we emphasize that deterministic Dice/IoU can penalize valid diversity (e.g., when a predicted plausible ``nodule'' mask is compared against an empty ground-truth annotation). This is precisely why GED remains the preferred metric for evaluating ambiguous segmentation tasks.

---

### Author Response · Authors · 2026-02-20
**Regarding Update: Additional Cross-Modality Results on a Second Multi-Annotator Dataset**

Dear AE and Reviewers,

We would like to share an important update addressing the concern regarding generalizability beyond LIDC-IDRI. Since submitting our rebuttal, we obtained access to an additional publicly available multi-annotator medical segmentation dataset (**IMA++ skin dermoscopy**), which contains 4–7 independent expert annotations per image. This dataset provides a substantially different modality (color dermoscopy instead of grayscale CT) and anatomical focus, making it a strong test of cross-domain robustness.

We trained and evaluated **AmbiguousTextDiff** on this dataset using the same framework (text-guided diffusion + ambiguity-aware hybrid objective) and identical ambiguity-aware evaluation metrics. The results confirm that our method generalizes effectively across modalities and continues to outperform probabilistic baselines on distributional metrics that measure both diversity and accuracy.

We will update the manuscript to include:
* A new quantitative results table for the second dataset,
* Qualitative figures showing diverse segmentation samples vs. multi-annotator ground truth,
* A detailed description of dataset preprocessing, prompt curation, and experimental protocol.

#### Quantitative Comparison on IMA++ (Skin Dermoscopy)

| Method | GED $\downarrow$ | CI $\uparrow$ | $D_{\text{max}}$ $\uparrow$ | $S_c$ $\uparrow$ | DA $\uparrow$ |
| :--- | :---: | :---: | :---: | :---: | :---: |
| Probabilistic U-Net | 0.2923 | 0.7713 | 0.8498 | 0.8488 | **0.9283** |
| PHI-Seg | 0.1950 | 0.7830 | 0.8405 | 0.9056 | 0.8966 |
| Generalized Prob. U-Net | 0.3371 | 0.7067 | 0.8269 | 0.7782 | 0.9033 |
| CIMD | 0.1389 | 0.7955 | 0.8988 | 0.9254 | 0.8578 |
| **AmbiguousTextDiff (Ours)** | **0.0724** | **0.8848** | **0.9510** | **0.9974** | 0.7538 |

These results strengthen our claim that text-guided diffusion with explicit ambiguity modeling is not restricted to lung CT nodules and remains effective in a distinct medical imaging domain with different ambiguity characteristics (e.g., low-contrast lesion boundaries, artifacts such as hair, and annotator style variability).

We appreciate the reviewers’ feedback that motivated this extension and will upload an updated manuscript revision incorporating these additional experiments and figures shortly.

Thank you,

Authors

---

> ### Author Response · Authors · 2026-02-20
> **Updated the paper manuscript**
>
> Dear AE and Reviewers,
>
> We would like to inform you that we have uploaded a revised manuscript incorporating all requested clarifications and additional experiments.
>
> The revision includes a new cross-domain evaluation on the IMA++ multi-annotator skin dermoscopy dataset, expanded quantitative and qualitative results, additional standard segmentation metrics (Dice/IoU), clarification of KL–MSE loss balancing and convergence behavior, and discussion of inference speed (DDPM vs. DDIM).
>
> We believe these updates substantially strengthen the manuscript, particularly regarding generalization and evaluation transparency.
>
> Thank you,
> Authors

---

### Decision · Action_Editor_HtBR · 2026-02-25

**Recommendation:** Reject

**Audience:**

Yes

**Audience Explanation:**

The paper's merits are appreciated by all reviewers.

**Claims And Evidence:**

No

**Claims Explanation:**

This work aims to address the inherent ambiguity in medical image segmentation. To this end, the authors introduce AmbiguousTextDiff, a novel text-guided diffusion framework, where the authors integrate semantic text prompts (curated from LIDC-IDRI dataset metadata, e.g., nodule texture, malignancy) with ambiguity-aware training, enabling the model to generate diverse. Extensive experiments on the full LIDC-IDRI test set are conducted to demonstrate the effectiveness of the proposed method.

**Evidence and support**

The primary limitations, as noted by most reviewers in their feedback and recommendations, center on the experimental validations. For example, one reviewer specifically suggested that

> The work is exclusively validated on lung nodule segmentation in CT scans (LIDC-IDRI). While LIDC-IDRI is a gold standard for ambiguity evaluation, the lack of experiments on other anatomical regions or imaging modalities raises questions about the method’s broader applicability. Clinical practice requires segmentation solutions across diverse organs and modalities, many of which have distinct ambiguity sources and limited structured metadata.

> The model employs pixel-space diffusion instead of latent diffusion, which is computationally expensive for high-resolution or 3D medical data. The authors do not report key efficiency metrics or compare with latent diffusion-based alternatives. For the method to be clinically viable, scalability to large-scale, high-dimensional data is critical—this limitation undermines its practical impact.

> While the paper emphasizes clinical relevance, the evaluation relies primarily on quantitative metrics and indirect qualitative comparisons with expert annotations. There is no formal validation by board-certified radiologists to assess."

The authors have recently revised their rebuttal, appearing to address certain previously raised concerns.

> important update addressing the concern regarding generalizability beyond LIDC-IDRI. Since submitting our rebuttal, we obtained access to an additional publicly available multi-annotator medical segmentation dataset (IMA++ skin dermoscopy), which contains 4–7 independent expert annotations per image. This dataset provides a substantially different modality (color dermoscopy instead of grayscale CT) and anatomical focus, making it a strong test of cross-domain robustness.

> We trained and evaluated AmbiguousTextDiff on this dataset using the same framework (text-guided diffusion + ambiguity-aware hybrid objective) and identical ambiguity-aware evaluation metrics. The results confirm that our method generalizes effectively across modalities and continues to outperform probabilistic baselines on distributional metrics that measure both diversity and accuracy.

*While I acknowledge the authors' recent efforts to address the limitations, the extent of the new experiments and the necessary related discussion suggest this constitutes a major revision suitable for a subsequent review round, rather than a minor revision. Typically, a minor revision involves only slight additions, considering the current experiments and post-validation results. Therefore, a new submission would be the more appropriate.*

**Resubmission Of Major Revision:**

The authors may consider submitting a major revision at a later time.